# The Importance of Legislative Reform to Enable Adaptive Management of Water Resources in a Drying Climate

Simone Stewart * and Graham Green

Department for Environment and Water, Adelaide 5000, Australia; graham.green@sa.gov.au
* Correspondence: simone.stewart@sa.gov.au; Tel.: +61-08-8463-6969

**Abstract:** In South Australia's Eyre Peninsula, groundwater provides 85% of the region's reticulated water supply. Fresh groundwater resides within shallow karstic limestone aquifers recharged by incident rainfall. Water levels are very responsive to short-term climate variability and are at risk of sustained decline due to long-term drying trends and the further rainfall declines indicated by projections of future climate, thereby increasing risk to water security and groundwater-dependent ecosystems. In 2009, a new adaptive resource management approach was enabled through legislative reform that better addresses climate variability, particularly where aquifer robustness is low. This allows the volume of water available for licensed allocations to be varied annually depending on the current condition of the aquifer resources. A three-tiered trigger level policy varies the rate at which water allocations are limited in proportion to monitored changes in groundwater storage. The three trigger thresholds are specified for each discrete groundwater resource, based on levels of risk. We now have more than five years of observations and practice of this approach to learn of its efficacy and consequences for water users, the water resources, and the environment. It has proved to be an effective way to deal with the uncertainties in how and when climate may change and how water management principles can effectively respond. Our case study provides an example of the importance of legislative reform to enable adaptive water resource management to effectively tackle the challenges of water planning in a drying climate.

**Keywords:** groundwater management; 3-D hydrostratigraphic modelling; adaptive management; Arc Hydro Groundwater; water allocation plan; karst aquifers; legislative reform

## 1. Introduction

South Australia is the driest state in the driest inhabited continent in the world. The majority of the state is characterised by an arid to semi-arid climate, with 96% of the state receiving less than 500 mm of rain each year [1]. In 2019, area-averaged annual rainfall for the state was less than 100 mm for the first time on record.

In many of the state's lower-population centres and rural areas, such as the Eyre Peninsula, water for potable, agricultural, recreational and industrial supply is sourced directly from local groundwater resources. Water allocation plans provide the framework for sustainable management of these resources by taking into account the competing environmental, social and economic demands for groundwater. In South Australia, a water allocation plan is a legal document which sets out the rules for managing the take and use of a prescribed water resource and is developed by a regional resource management authority under the *Landscape South Australia Act 2019* (the Act), with input from the community, industry and key stakeholders.

A groundwater resource is prescribed by regulation when the water use in an area is at a level at which regulatory control is needed to ensure a sustainable rate of extraction and to maintain water dependent ecosystems. Once a resource is prescribed, the taking of water from the resource (with some exclusions) requires a licence from the relevant authority [2]. A water allocation plan ensures that the needs of the environment are considered when

determining how much water is made available for consumptive purposes such as public water supply, irrigation, industrial processes, and stock watering. Each water allocation plan is reviewed within 10 years of its adoption to ensure the management principles are effective and the resource is being managed sustainably in light of the latest available science and monitoring data.

Groundwater resources that are particularly sensitive to water stresses and that rapidly respond to changes in recharge, such as those on the Eyre Peninsula, can benefit from an adaptive management approach. A water allocation plan enables the volume of water allowed to be taken to be restricted in periods of below average recharge in order to preserve the water source for use in future years. This approach also protects groundwater dependent ecosystems and ensures the maintenance of continued freshwater discharges to estuarine and near-shore marine environments. Within the areas discussed through this case study, there are wetlands of national importance and other valued ecosystems that are dependent on and sensitive to groundwater levels. These need to be managed and protected appropriately through the management mechanism.

The experience of using climate-sensitive groundwater resources as the only source water for the local community is not unique to the Eyre Peninsula. Harrington and Cook [3] identified 14 priority aquifers which were both sensitive to climate change and regionally important, including the contrasting landscapes of the Otway Basin in the South East of South Australia, and the Atherton Tablelands in Far North Queensland.

Further afield, Cuthbert et al. [4] identified that the shortest groundwater response times to recharge are generally observed in wet, humid regions, such the Amazon, the Congo basin and Indonesia, as well as low-lying regions, such as the Asian mega-deltas and the Florida Everglades making these particularly vulnerable to changes in rainfall recharge that may occur due to a changing climate. Additionally, it was identified that aquifers in Europe and North America could also respond rapidly and be sensitive to water stresses, such as the chalk aquifer in the Thames Basin, which provides freshwater to much of southern and eastern England [4].

There have been two recent reviews of water management plans in Australia [5,6]. The first tested groundwater management plans using a control theory approach. Seven components of a management plan equivalent to basic components of a control loop were determined, and requirements of each component necessary to enable testability were defined. A defined objective or acceptable impact was necessary for plans to be testable. When applied to 15 Australian groundwater management plans, approximately 47% were found to be testable.

In the second, groundwater management case studies were critically evaluated with reference to adaptive management principles. They identified substantial variability in the interpretation of adaptive management principles across eleven groundwater case studies. Comparison of published adaptive management guidelines and groundwater examples of adaptive management plans revealed only a small number of those management plans adhered to the key components of adaptive management guidelines. The most notable issues in the application of adaptive management to groundwater activities include a lack of substantive mitigation measures and/or assessment of the potential for remediation. Both of these studies suggest that a more rigid approach with regards to specifying the mitigation measures within groundwater management plans is required, with many so-called adaptive management plans having poorly defined measures for the mitigation of potential negative impacts.

This paper presents a case study on adaptive water resource management in situations where a community is exclusively dependent on a groundwater resource with small storage volumes that are highly responsive to rainfall and therefore subject to decline with relatively moderate but sustained changes in rainfall in the context of a semi-arid climate.

The management approach discussed in this paper was enabled through detailed knowledge of the aquifer extents and characteristics, resulting from extensive drilling and geophysical surveys. To apply a similar approach in other locations, the appropriate aquifer knowledge

would be required. Additionally, legislation with the appropriate provisions is required that allows for adaptive management in response to changes in the condition of the resource.

The role of legislation in enabling adaptable water policies that can respond to climate change was examined by Nanni [7], identifying a trend in water legislation towards acknowledging the need to provide the flexibility required for addressing climate issues.

While adaptive management of water resources through the use of triggers is not a new concept either worldwide or for the management of other South Australian water resources, the approach discussed in this paper provides an example of how to undertake trigger level management for a data rich, climate responsive, unconfined aquifer system which is experiencing a climate consistent with the climate change projections.

South Australia has 23 inland prescribed areas where either the wells, watercourses, surface water or some combination of these are prescribed in certain areas of the state (Figure 1). Of these prescribed areas, 10 have an adaptive management approach built into the water allocation plan. The robustness of the adaptive management approach varies greatly, however. Water allocation plans which were developed prior to the amendments in the legislation (further discussed in Section 2.3 of this paper), or which were already well underway when the legislative reform took place, were somewhat limited in the extent to which they could provide for adaptive management due to the restrictions in the legislation at the time.

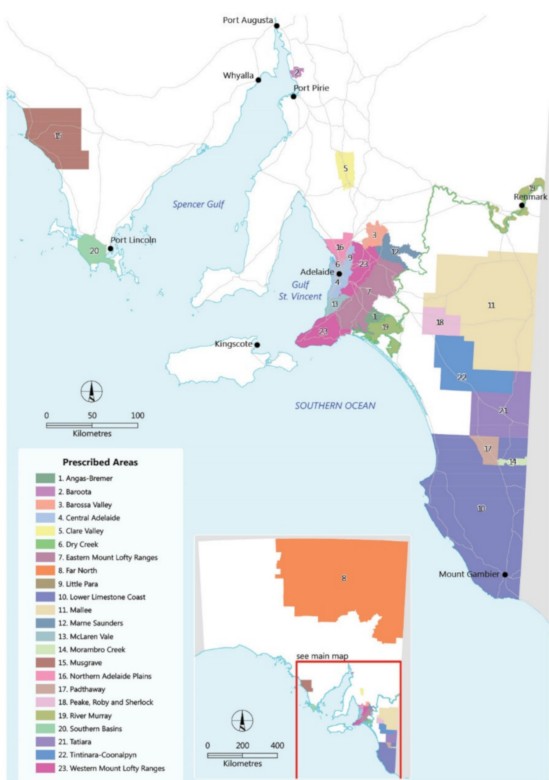

**Figure 1.** Prescribed areas of South Australia.

Adaptive management in a South Australian context generally falls into three categories, they include:

- Review: For over-allocated resources with reduction schedules built into the water allocation plan, the management approach includes a review point at which it was considered whether the reduction schedule was required to be maintained or could be discontinued based on how the resource was tracking with regard to groundwater level declines or salinity increases [8–11].
- Investigation: In other resources, a breach of a monitoring trigger initiates an investigation into the cause of the breach, assessment of the options for remediating the

breach (including applying changes to allocations or the water taking regime) and implementing those remedial actions [12–14].

- Action: Triggers are utilised to vary the volume of water available for allocation in periods of lower water security or when resources are declining to critical thresholds [15–17].

The above provides examples of how trigger levels are utilised in prescribed resources across South Australia to enact adaptive resource management. With the exception of the River Murray, the Eyre Peninsula is the only other resource in the state adaptively managed by taking into account the water available in storage to determine the volumes of water which can be allocated annually in response to changes in recharge to the resource.

While there are some adaptive groundwater management plans within Australia which do provide a prescriptive mechanism by which allocations may be changed based on resource condition [18,19], similar to the approach discussed in this paper, these are generally based on a monitoring of groundwater levels and/or groundwater salinity (in relation to monitoring sea water intrusion) rather than considering aquifer storage as a mechanism for change.

## 2. Background

### 2.1. History of Water Allocation Plans for the Southern Basins and Musgrave Prescribed Wells Areas

The Southern Basins and Musgrave Prescribed Wells Areas were prescribed in 1987 to better manage and protect groundwater resources used for the Eyre Peninsula's public water supplies [20]. In 1997, the first water management document was prepared for the prescribed wells areas of the Eyre Peninsula, this being the County Musgrave and Southern Basins Water Resources Management Objectives and Policies 1997 [21]. Subsequently, water allocation plans were developed for the Southern Basins and Musgrave Prescribed Wells Areas, to regulate the use of the groundwater resources.

The inaugural Water Allocation Plans for the Southern Basins Prescribed Wells Area and the Musgrave Prescribed Wells Area were adopted on 31 December 2000 and 2 January 2001, respectively [22,23]. The regional resource management authority reviewed these water allocation plans in accordance with the requirements of the Act and subsequently made the decision to amend both plans due to their inability to respond appropriately to the changing climate. They also took the opportunity to combine them into a single plan through this process.

Changes to the *Natural Resources Management Act 2004*, the water management legislation in South Australia at the time, were enacted in 2009 and enabled a more suitable management approach for the highly climate responsive resources within these prescribed wells areas. In 2016, a water allocation plan for both the Southern Basins and Musgrave Prescribed Wells Areas incorporated this adaptive management approach and was subsequently adopted by the then Minister for Sustainability, Environment and Conservation [17]. The implementation of this management approach since 2016 has provided an evidential case study of the effectiveness and limitations of such an approach in the context of a highly variable water resource in an area with declining average rainfall.

### 2.2. Hydrogeology

The Southern Basins and Musgrave Prescribed Wells Areas are located on Eyre Peninsula in South Australia, approximately 300 km west of Adelaide (Figures 2 and 3). There are limited surface water resources because of the permeable nature of the dune landscape, which readily absorbs most of the rainfall, and the supply of good quality groundwater is limited. Groundwater is the principal source of water for town water supply, irrigation, and stock and domestic purposes, with the groundwater resources found in the Quaternary Limestone aquifers across the prescribed wells areas being vital for the security of Eyre Peninsula's water supply.

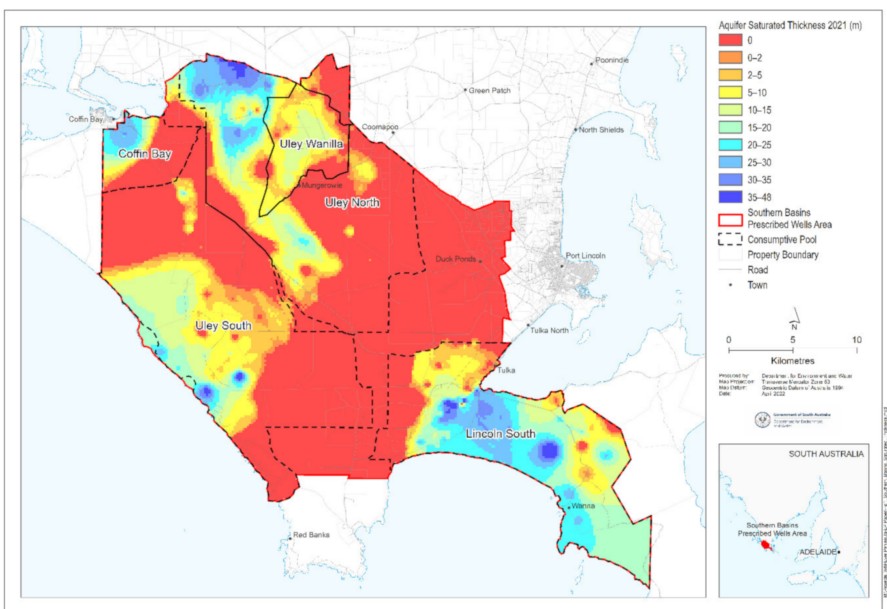

**Figure 2.** Location of the Southern Basins Prescribed Wells Areas, the consumptive pools, and the saturated thickness of the Quaternary Limestone aquifer in 2021.

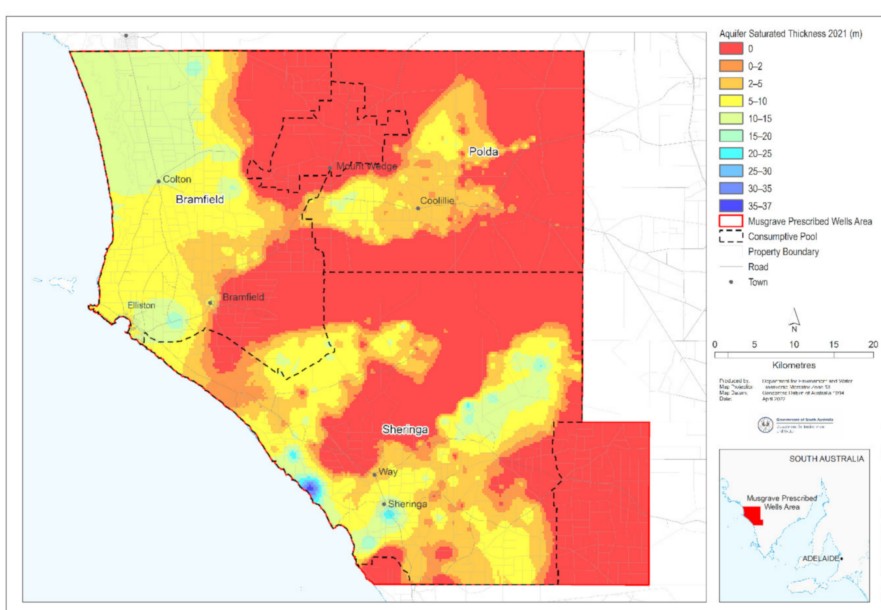

**Figure 3.** Location of the Musgrave Prescribed Wells Areas, the consumptive pools, and the saturated thickness of the Quaternary Limestone aquifer in 2021.

Potable groundwater is found mainly within the Quaternary Limestone aquifer, which forms a generally thin veneer over the older Tertiary sediments and is continuous across both prescribed wells areas. The limestone consists of sand-sized shell fragments, calcareous algae fragments and quartz grains that were deposited as large barrier dunes on ancient shorelines during high sea level stands in the Pleistocene era. These sediments are known to be over 130 m thick in parts of the Uley South area [24]; however, they are generally much thinner across the remainder of the prescribed wells areas.

The saturated thickness of the Quaternary Limestone aquifer can fluctuate rapidly in response to rainfall. With the exception of a few specific locations, the saturated thickness of the aquifer is generally less than 20 m for the Southern Basins Prescribed Wells Area

(Figure 2) and 10 m for the Musgrave Prescribed Wells Area (Figure 3), with large areas of the prescribed wells areas being unsaturated.

The saturated areas within the Quaternary Limestone aquifer are largely dependent on local rainfall falling on the overlying land for recharge. There are no regional inflows of either surface water or groundwater to the region's groundwater systems [25]. Analyses of water table fluctuations within the Quaternary Limestone aquifer show that recharge occurs only after intense rainfall events [26]. This evidence suggests that short-lived runoff allows water to percolate through dissolution features (sink holes) and reach the water table rapidly. Investigations of the Polda Basin system, located within the Musgrave Prescribed Wells Area (Figure 3) indicate that recharge only occurs when the area receives more than 60 mm of rainfall in a month between the months of May and October [26]. The Uley Basin system, located within the Southern Basins Prescribed Wells Area (Figure 2), shows recharge only when the Quaternary Limestone aquifer in the region receive more than 10 days of greater than 10 mm of rainfall between the months of May and October [24]. However, hydrochemical and isotopic data has since been used to infer the nature of recharge pathways and evapotranspiration processes [27]. It was concluded that sinkholes may act to bypass the shallow soil zone and redistribute infiltrating rainfall into the deeper unsaturated zone, rather than acting as conduits between the ground surface and the water table.

The vulnerability of these aquifers to sustained rainfall changes was investigated and numerical models of groundwater recharge and surface water runoff were developed for the target water resources and calibrated against historic groundwater level and flow data to ensure the models appropriately represented the variability of key hydrological records in response to annual variations in key climate variables [28]. A relationship between future reductions in annual rainfall as a result of climate change and reductions in both surface runoff and groundwater recharge was determined and used to summarise the potential impacts on surface water and groundwater of the median climate change projections available at that time. In the Southern Basins Prescribed Wells Area, the simulated reductions in groundwater recharge resulting from median climate scenarios projected by CSIRO and BoM [29] ranged from 11% in a 2030 climate to 47% in a 2070 climate. The projected decline in rainfall in this region is consistent with climate projections reported in the 2022 report of IPCC Working Group II [30], which projects a decline in winter and spring rainfall and higher evaporation rates over southern mainland Australia.

Due to the responsiveness of these systems to variations in rainfall, determining a sustainable extraction limit for the groundwater resources is extremely difficult. To enable localised management, each prescribed wells area was separated into multiple consumptive pools (Figures 2 and 3), similar to a management zone but within which the water to be used for consumptive purposes is accounted for and managed in a manner which responds to the specific issues within the consumptive pool. Such as considering the presence of groundwater dependent ecosystems, the presence of alternative water supplies or the potential impacts of saltwater up-coning or intrusion when determining how to allocate groundwater.

To demonstrate the correlation between rainfall and recharge, Figure 4 displays hydrographs for four wells within the Sheringa consumptive pool (identified in blues—see Figure 5 for location) where there is no licensed extraction occurring, alongside four hydrographs for a well located in the Bramfield consumptive pool (identified in greens—see Figure 5 for location), within which extraction occurs for irrigation and provides for public water supply. The cumulative deviation from average annual rainfall graphed in yellow identifies periods where rainfall trends are above or below average. An upward slope indicates a period where the rainfall is above average, while a downward slope indicates a period where the rainfall is below average. Groundwater levels are highly correlated with trends in rainfall. Where there is a period of below average rainfall, groundwater levels decline, and in periods of above average rainfall, such as in 2009 and 2010, groundwater levels recover. All consumptive pools, irrespective of extraction occurring, show significantly similar trends indicating that recharge is a major driver of the system. Given

the water allocation plan can only manage the extractions of water and has no ability to manage the recharge variability to the system, it is the limitations to extractions which enables the adaptive management of these resources to occur.

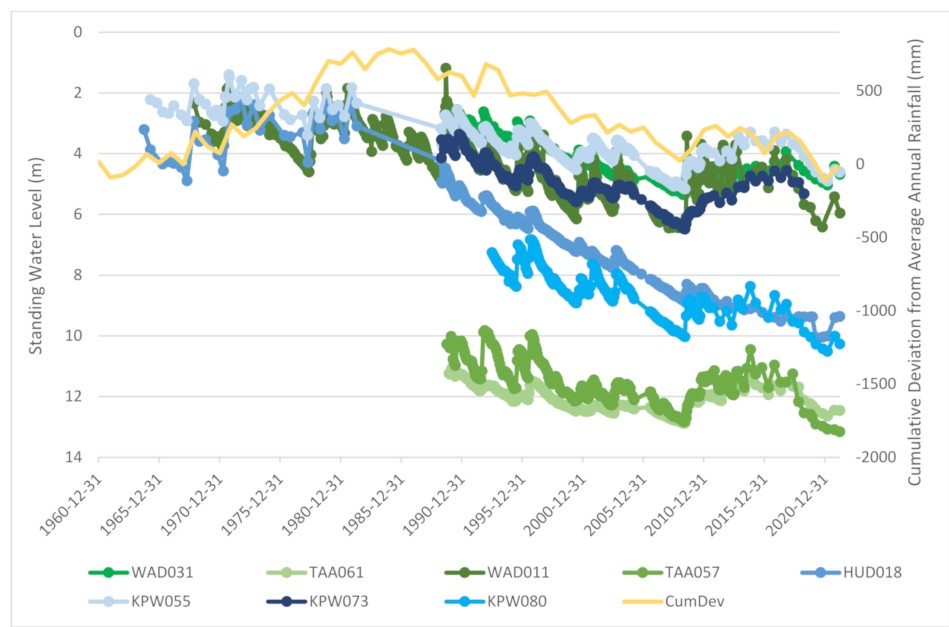

**Figure 4.** Groundwater level trends of the Sheringa and Bramfield consumptive pools of the Musgrave Prescribed Wells Area and the rainfall trend.

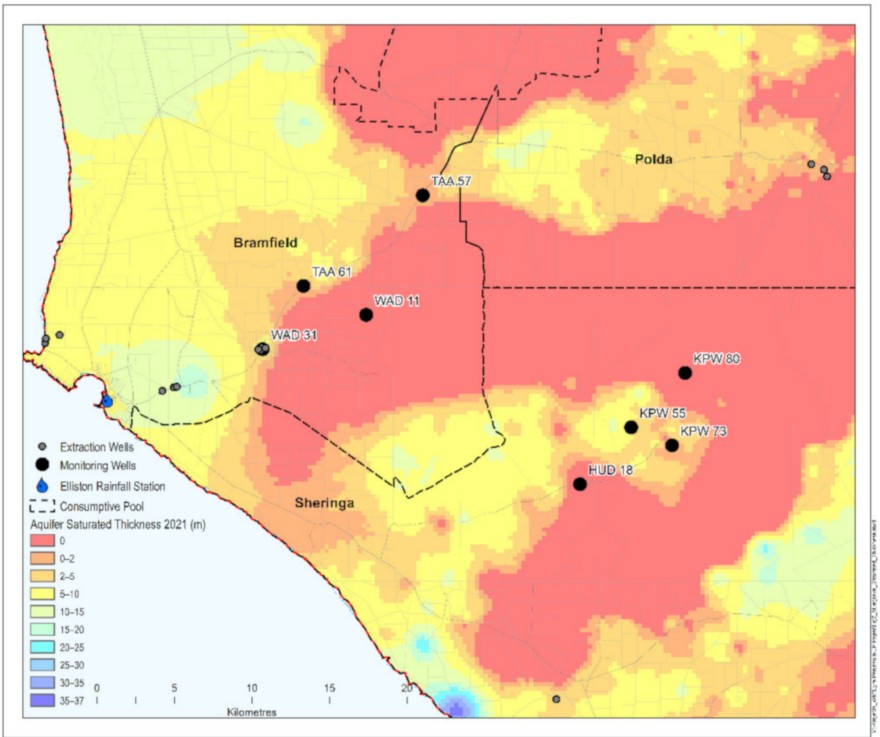

**Figure 5.** Location of monitoring wells for groundwater level trends of the Sheringa and Bramfield consumptive pools of the Musgrave Prescribed Wells Area and the location of the Elliston rainfall station.

*2.3. Management Framework*

In view of the highly variable nature of these groundwater resources due to their strong reliance on contemporary rainfall, it is appropriate to have water allocations which vary from one year to the next based on observations of the resource condition. This was the case to an extent in previous water allocation plans for the two prescribed well areas on the Eyre Peninsula, which used a 10-year rolling average of recharge estimates to determine the volumes of water to be allocated annually. However, this process failed to take into account water storage levels in the aquifers and the policy of responding to a 10-year rolling average of recharge resulted in an overly-delayed response to declining aquifer levels as rainfall declined in the first years of the Millennium Drought (the longest uninterrupted series of years with below median rainfall in southeast Australia since at least 1900, being the period 2001–2009).

The legislation at the time did not expressly allow for a water allocation plan to vary allocations without also varying licensees' ongoing rights to access water. This was highly problematic to water licensees, as their allocation was a commodified water right and each time an allocation was reduced, their water right became less valuable or mortgageable to a bank for security of business loans.

A legal mechanism by which allocations could be varied annually in response to the strong climatic and groundwater level variability observed on the peninsula, which did not impact licensees' water rights, was required.

In 2004 South Australia became a signatory to the *Intergovernmental Agreement on a National Water Initiative* [31]. The National Water Initiative is an agreement signed by state and territory governments and the Australian Government. It sets out principles on which freshwater resources should be shared for the benefit of communities, freshwater ecosystems and economic development. These principles relate to matters such as the need for science-based water planning, adaptive management of the resource, open engagement with communities, secure water rights for consumptive purposes, and the provision of environmental water requirements. A key aspect of the National Water Initiative is that it has provisions guiding the states to implement a new planning framework which resulted in the ongoing right to access water (water access entitlement) being a separate management authorisation to the volume of water which can actually be taken annually (the allocation).

As a signatory to the National Water Initiative, the South Australian Government amended the then *Natural Resources Management Act 2004* (now replaced by *Landscape South Australia Act 2019*) in 2007 to create the system for separating an existing water licence into separate water management authorisations (unbundling the licence). The amendments came into force on 1 July 2009, with transitional arrangements put in place so that water rights for prescribed water resources were not required to be unbundled until the relevant water allocation plan was amended.

This legislative reform provided the opportunity for the licences issued for the Southern Basins and Musgrave Prescribed Wells Areas to be unbundled upon adoption of a new water allocation plan, and for the water allocation plan to enact an adaptive management framework for the highly responsive resources. This enabled annual allocations to be reduced in the event that minimal recharge to the aquifers occurred and declines in groundwater level were observed, without impacting on licensees' longer-term water rights.

In an unbundled water planning framework, water allocations are obtained through a water access entitlement, which is the ongoing right to a share of the resources. It is a privately owned asset of the licensee, which can be mortgaged, sold, or temporarily transferred in part or in full. A water allocation is the volume of water that may be taken during a particular water use year (1 July to 30 June). The water allocation is determined based on the value of the entitlement share where the value of the share is subject to variation in relation to the condition of the resource.

## 3. A Revised Water Allocation Methodology

In the 2016 Water Allocation Plan for the Southern Basins and Musgrave Prescribed Wells Areas, determining the value of the entitlement share is achieved through analysing the change in storage (or aquifer saturated thickness) on an annual basis using the latest monitoring data in combination with the Aquaveo™ Arc Hydro Groundwater model (Aquaveo, UT, USA). The assessment of the level of storage (expressed as a percentage compared to a baseline) for each resource (consumptive pool) determines any changes to allocations for the next water use year.

The baseline was required as a means to set triggers for change in the volume of water available for allocation, and to identify a storage trigger which would result in the complete cessation of extraction when the resource sustainability was threatened. Choosing a baseline period when groundwater levels were at their historical high was not appropriate, as it did not take into account the recent climatic conditions, which had generally been drier than that observed historically. The period 1992 to 2012 provided a reasonable period over which climatic conditions were considered to be contemporary. Consequently, the baseline for the current climate was set as the year in which groundwater levels were at their highest since 1992. Groundwater level monitoring data indicated this generally occurred for most wells across the prescribed wells areas in 1993. The saturated thickness of the aquifer for April (autumn) of 1993 was defined as the aquifer storage baseline. April was chosen, as it represents the time of year when groundwater levels were likely to be at their lowest elevation after summer and the irrigation season, but prior to any significant recharge occurring. This is a precautionary approach which ensures water is not allocated without taking into account the seasonal declines that are observed due to the natural discharge from aquifers.

The storage level for each resource is calculated annually using the latest monitoring data from a specific set of monitoring wells. The data for April of the relevant year is interpolated to generate a layer which represents the groundwater level in relation to sea level (reduced standing water level). This layer is overlain on the Tertiary Clay layer, which is based on the lithological logs available for each prescribed wells area and outlines the depth at which the Tertiary Clay is intercepted, it therefore outlines the base of the Quaternary aquifer. As lithology will not change, this layer is fixed for the life of the water allocation plan. The ArcHydro Groundwater software package was then utilised to model the thickness of the water within each resource, from this model a void volume of the storage component is calculated. It should be noted that the ArcHydro model is not a numerical model which accounts for density driven flow, it is a 3-D hydrostratigraphic structural model which can provide for the calculations of storage within the aquifer. The volume of storage for each resource for the relevant year is compared to the volume storage for each resource for the baseline storage level of 1993. As the baseline level of storage is taken to equate to 100%, the difference from the baseline is calculated as a percentage for the relevant year.

Annual allocations are managed through a proportional relationship between the level of storage in the aquifer and the proportion of water available from the resource for allocation. The proportional relationship has fixed triggers which define the degree of variation to water allocations that will occur. These triggers are:

- The Upper Storage Trigger: when the water available from a resource for allocation falls below 100% of that outlined as available for allocation in the Water Allocation Plan
- The Mid Storage Trigger: when a variation to the Rate of Change to the percentage of water available, as a function of the rate of change in the assessed level of storage, occurs
- The Lower Storage Trigger: when the water available from a resource for allocation falls to 0% of that outlined as available for allocation in the Water Allocation Plan

When the storage level for a resource is greater than the Upper Storage Trigger, the resource is considered to be a low risk and the water available for allocation is 100% of the licensee's water access entitlement. When the storage level falls below the Upper Storage

Trigger but remains higher than the Mid Storage Trigger, the resource is considered to be at low–moderate risk and the volume of water available for allocation varies by the Upper Rate of Change for each 1% change in the aquifer storage level. When the storage level falls below the Mid Storage Trigger but remains higher than the Lower Storage Trigger the resources is considered to be at moderate–high risk and the volume of water available for allocation varies by the Lower Rate of Change for each 1% change in the aquifer storage level. When the storage level is assessed to be equal to or less than the Lower Storage Trigger the resource is considered to be at high risk and as such no water is available from the resource for allocation. The rate of change is defined as the rate at which the share of the resource (allocation) that the licensee is entitled access to, changes with the level of storage.

The development of the defined triggers (Table 1) took into account the specific characteristics of the individual resources, the availability of alternative water supplies and the needs of any ecosystems dependent on the groundwater resource.

**Table 1.** Trigger levels and rates of change for the various resources.

| Prescribed Wells Area | Consumptive Pool | Upper Storage Trigger % | Upper Rate of Change | Mid Storage Trigger % | Lower Rate of Change | Lower Storage Trigger % |
|---|---|---|---|---|---|---|
| Southern Basins | Coffin Bay | 95 | 1.0 | 94 | 49.5 | 92 |
| | Uley Wanilla | 85 | 1.0 | 79 | 10.11 | 70 |
| | Uley North | 90 | 0.5 | 86 | 24.5 | 82 |
| | Uley South | 90 | 1.0 | 81 | 9.1 | 71 |
| | Lincoln South | 95 | 1.0 | 94 | 99 | 93 |
| Musgrave | Polda | 100 | 1.0 | 84 | 5.25 | 68 |
| | Bramfield | 90 | 1.0 | 81 | 10.11 | 71 |
| | Sheringa | 100 | 0.5 | 87 | 6.68 | 73 |

The scientific report [32] upon which the water allocation plan is based describes the methodology for determining the triggers and rates of change for the varied resources of the Quaternary Limestone aquifer in detail; however, the following sections provide an overview of how the storage triggers and rates of change were determined for the consumptive pools

### 3.1. Upper Storage Trigger

The Upper Storage Trigger was determined by considering the vulnerability of each consumptive pool. Consumptive pools which were more variable had a greater susceptibility to rapid change or falling below critical levels with little warning. The likely degree of variability in the storage levels in each resource was able to be assessed throughout a scenario testing period which, using historic monitoring data, calculated the storage for each consumptive pool for the period 2000 to 2012.

If the storage level variability was high (the level of storage modelled over the scenario testing period varied by more than 20%), the Upper Storage Trigger was set at 100%, indicating that as soon as the saturated thickness falls below the baseline value, allocations will vary by the Upper Rate of Change. This prevents the need to impose significant reductions on licensees' allocations in one year and allows forward planning for licensees and time to source alternative water resources.

If the scenario testing indicated that the storage level variability was moderate (the level of storage modelled over the scenario testing period varied between 10–20%), the Upper Storage Trigger was set at 90%, thereby allowing minor declines in storage before allocations needed to be adjusted.

If the scenario testing indicated that the variability was low (the level of storage modelled over the scenario testing period varied by less than 10%), the Upper Storage Trigger was set at 85%.

Exceptions to the above are the coastal consumptive pools where up-coning or saltwater intrusion were a risk. In the Coffin Bay and Lincoln South consumptive pools up-coning

is a significant risk to the resource. In these consumptive pools, the fresh water portion of the aquifer is quite thin due to the stratification on saline water. Due to the connection with the ocean, these resources are buffered from variability in the level of storage observed over the scenario testing period. However, in order to minimise the risk of saltwater up-coning, in these cases, the Upper Storage Trigger was set at 95% of the baseline level.

In the Uley South consumptive pool, it is not saltwater up-coning that poses a risk to the resource but rather, saltwater intrusion. Saltwater intrusion is the lateral movement of saline water into a coastal aquifer. Monitoring of the saltwater interface has been undertaken in the Uley South consumptive pool and to date no movement of the interface has been observed. However, as potential saltwater intrusion is a risk to the resource, it was taken into account when assigning triggers for the consumptive pool. Due to the connection to the ocean, the variability in the level of storage over the scenario testing period for the Uley South consumptive pool was low; however due to the risk of saltwater intrusion, a higher Upper Storage Trigger of 90% has been set in an attempt to mitigate this risk.

### 3.2. Mid Storage Trigger

The Mid Storage Trigger was set as the median value between the Upper and Lower Storage Trigger values.

### 3.3. Lower Storage Trigger

The Lower Storage Trigger was determined via two mechanisms dependent on the location of the consumptive pool within the prescribed wells area: inland consumptive pools (being the Uley Wanilla, Uley North, Polda, Bramfield and Sheringa consumptive pools) or coastal consumptive pools (being the Uley South, Coffin Bay and Lincoln South consumptive pools).

#### 3.3.1. Inland Consumptive Pools

The Lower Storage Trigger for the inland consumptive pools was based on an assessment of the aquifer robustness, which is the ratio between aquifer storage and recharge. In aquifers with a large robustness (large storage compared to recharge) there may be opportunities to use the aquifer storage to buffer natural variations in climate [33]. While threshold values which define high and low robustness have not been comprehensively agreed upon, based on case studies [33], it is suggested that an index of 100 or higher indicates high robustness, while an index of less than 20 indicates a low robustness.

For those consumptive pools identified as having a low robustness (Uley North, Bramfield, Polda and Sheringa), the Lower Storage Trigger was set to be equivalent to the level of storage which corresponded to the minimum water levels observed over the scenario testing period (2000–2012), being the peak of the Millennium Drought, either 2008 or 2009 depending on the consumptive pool. However, given the aquifer geometries for the Polda consumptive pool the values of the minimum water level observed over the scenario testing period were likely too low to maintain the aquifer on an ongoing basis. As such the Lower Storage Trigger for the Polda consumptive pool was required to be further refined.

The Uley Wanilla consumptive pool was identified as having a high robustness and therefore it was considered that the Lower Storage Trigger could be lower than that minimum storage observed over the scenario testing period given the robustness index was 149.

For both the Polda and Uley Wanilla consumptive pools the Lower Storage Trigger was set to be equivalent to the depth at which the aquifer can no longer reasonably supply water for licensed purposes. The Thies Solution [34] provides a mechanism by which to calculate the maximum drawdown observed at 0.1 m from a pumping well for specific extraction volumes. Reasonable estimations of transmissivity and specific yields for the relevant consumptive pools were collated from the literature and historic aquifer tests and an appropriate pumping rate based on historical use from the consumptive pool was utilised to determine the likely drawdown which would be observed at 0.1 m from the well and the

drawdown likely to be observed within the well itself. By comparing this drawdown to the saturated thickness of the aquifer as measured in 2011 (when the triggers were being developed), and then comparing this to the level of storage for 2011, an equivalent level of storage which related to the minimum allowable saturated thickness was able to be determined which results in the Lower Storage Trigger for these consumptive pools.

### 3.3.2. Coastal Consumptive Pools

The Lower Storage Trigger for the coastal consumptive pools was determined by considering a critical minimum thickness of aquifer saturation which was based on mean sea level (0 m AHD). This is because within both the Coffin Bay and Lincoln South consumptive pools, the water is saline below this level and effectively cannot be used for the purposes of irrigation, public water supply or other licensed requirements. In Uley South, this is a precautionary measure to minimise the risk of saltwater intrusion.

### 3.4. Upper Rate of Change

The Upper Rate of Change was related to the accessibility risk to the groundwater of the consumptive pool as assessed in a risk assessment [35]. The accessibility risk was deemed to be the risk to users (licensed and non-licensed) of restricting water for consumptive use. That is, the risks to social, cultural and economic values of not having access to groundwater. Two types of risk issues were used to assess the accessibility risk for each consumptive pool:

1.  Users are not allocated sufficient groundwater to meet their consumptive needs.
2.  There is no alternate water supply available to meet user needs.

By considering the likelihood and consequence of the different risks, scores for individual consumptive pools were able to be determined ranking the accessibility risk as either High, Moderate or Low.

If the accessibility risk was high, the Upper Rate of Change was defined as a 1% change in allocation per 1% change in the storage level. This is because the Upper Rate of Change relates to the change in storage between the Upper Storage Trigger and the Mid Storage Trigger and within this zone, there is a low likelihood that the resource will be at risk if water is continued to be allocated. Therefore, it is considered acceptable to continue to allocate water in a reasonable manner to licensees and for public water supply purposes in these areas where alternative water supplies are not available. When the accessibility risk is low (indicating the resource is not highly essential for licensed purposes or alternative supplies are available) allocations are varied by 0.5% for every 1% of change in storage level.

### 3.5. Lower Rate of Change

The Lower Rate of Change is the rate of change required to ensure that for each 1% change in the level of storage between the Mid Storage Trigger and the Lower Storage Trigger, an appropriate corresponding rate of change is applied to the volume of water available for allocation, such that when it reaches the Lower Storage Trigger the volume of water available for allocation equals zero.

### 3.6. Annual Water Allocations

Under the water allocation plan, a licensee's water access entitlement equates to the volume of water which would be available under the baseline 1993 storage levels scenario (the best case), and annually the allocations are varied from this maximum volume based on the assessed level of storage within the resource for that year, thereby reflecting the condition of the resource in the water availability.

An example using three different levels of storage is shown in Figures 6 and 7 to demonstrate how the proportional relationship works for the Bramfield consumptive pool. Once the storage level for the resource is calculated, the percentage of storage is located on the 'level of storage' x axis and the corresponding 'proportion of water available

for allocation' is determined from the y axis by identifying where the purple line (the proportional relationship) is intercepted.

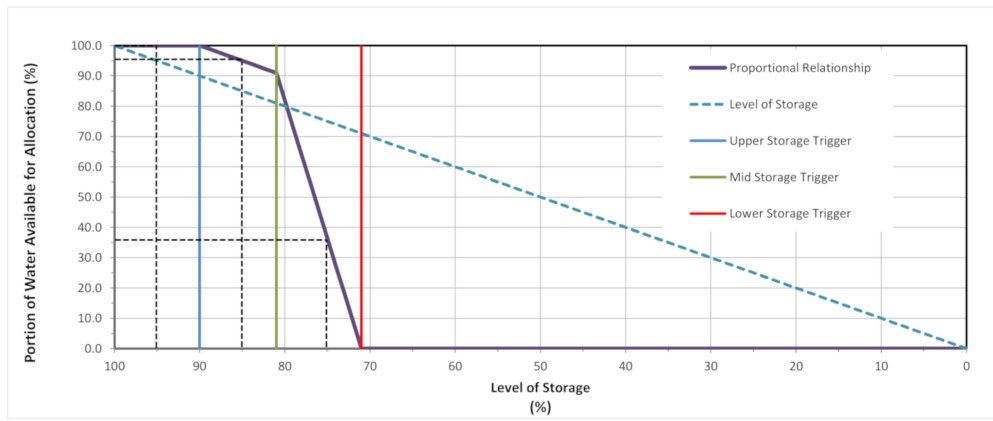

**Figure 6.** Proportional relationship and example diagram of allocations for the Bramfield consumptive pool.

Example 1

Level of Storage: 95%

Proportion of the
WAE allocated: 100%

Example 2

Level of Storage: 85%

Proportion of the
WAE allocated: 95%

Example 3

Level of Storage: 75%

Proportion of the
WAE allocated: 36.4%

**Figure 7.** Example calculations for allocations for the Bramfield consumptive pool.

This provides the approach for how the allocations for each consumptive pool are determined annually. Licensees are then required to ensure that they take water in accordance with their allocated volumes.

## 4. Results

The ability to amend the legislation in order to separate the property rights for water (water access entitlement) and the volume of water available (allocation) has enabled a management framework which is able to respond rapidly to resource condition changes, while maintaining licensee's water rights and ensuring the water requirements for groundwater dependent ecosystems are provided even in times of reduced water availability. Following the implementation of the Water Allocation Plan for the Southern Basins and Musgrave Prescribed Wells Areas, the process of managing groundwater resources to resource condition triggers in place of managing allocations within sustainable yield volumes is being

rolled out in other areas of the State such as the Northern Adelaide and Central Adelaide Plains Prescribed Wells Areas [16].

Implementation of the water allocation plan since 2016 has been responsive to the declining water availability. While the methodology devised above was tested to see how the allocations would have been varied throughout the Millennium Drought period, more rapid reductions in allocation have been required since the adoption of the water allocation plan given the significant dry periods experienced on the Eyre Peninsula since 2016, resulting in two resources being on zero allocations in 2021 (Table 2 and Table 3).

**Table 2.** Level of storage and allocations for each resource in the Southern Basins PWA.

| Water Use Year | Coffin Bay | | Uley Wanilla | | Uley North | | Uley South | | Lincoln South | |
|---|---|---|---|---|---|---|---|---|---|---|
| | Storage (%) | Allocation (%) | Storage (%) | Allocation (%) | Storage (%) | Allocation (%) | Storage (%) | Allocation (%) | Storage (%) | Allocation (%) |
| 2015–2016 | 99.31 | 100.0 | 87.46 | 99.0 | 87.48 | 98.5 | 93.34 | 100.0 | 95.56 | 100.0 |
| 2016–2017 | 98.80 | 100.0 | 85.28 | 97.0 | 83.59 | 49.0 | 92.24 | 100.0 | 95.15 | 100.0 |
| 2017–2018 | 99.02 | 100.0 | 84.59 | 97.0 | 83.09 | 24.5 | 91.05 | 100.0 | 95.84 | 100.0 |
| 2018–2019 | 99.31 | 100.0 | 83.41 | 95.0 | 81.34 | 0.0 | 90.16 | 100.0 | 95.52 | 100.0 |
| 2019–2020 | 98.83 | 100.0 | 81.43 | 93.0 | 79.62 | 0.0 | 88.78 | 99.0 | 95.02 | 100.0 |
| 2020–2021 | 98.57 | 100.0 | 80.66 | 93.0 | 77.85 | 0.0 | 87.07 | 97.0 | 94.72 | 100.0 |
| 2021–2022 | 98.00 | 100.0 | 79.00 | 91.0 | 77.00 | 0.0 | 86.84 | 97.0 | 94.00 | 99.0 |

**Table 3.** Level of storage and allocations for each resource in the Musgrave PWA.

| Water Use Year | Polda | | Bramfield | | Sheringa | |
|---|---|---|---|---|---|---|
| | Storage (%) | Allocation (%) | Storage (%) | Allocation (%) | Storage (%) | Allocation (%) |
| 2015–2016 | 78.84 | 57.8 | 85.66 | 95.7 | 88.77 | 94.4 |
| 2016–2017 | 74.25 | 31.5 | 81.88 | 92.0 | 86.17 | 86.8 |
| 2017–2018 | 78.76 | 57.8 | 84.01 | 94.0 | 88.75 | 94.5 |
| 2018–2019 | 69.92 | 10.5 | 84.10 | 94.0 | 87.66 | 94.0 |
| 2019–2020 | 64.07 | 0.0 | 78.05 | 63.7 | 84.84 | 80.1 |
| 2020–2021 | 57.28 | 0.0 | 74.51 | 31.9 | 79.26 | 40.1 |
| 2021–2022 | 52.88 | 0.0 | 72.38 | 12.6 | 75.92 | 20.0 |

The water allocation plan, and hence the process for allocating water as discussed in this paper, has been in place for the past six years. Figure 8 uses the Bramfield consumptive pool as an example to display how the saturated thickness of the aquifer has changed over this period in response to the changing rainfall and extraction levels. Figure 9 displays the groundwater level trends for the monitoring wells identified in Figure 8 for the period the water allocation plan has been active.

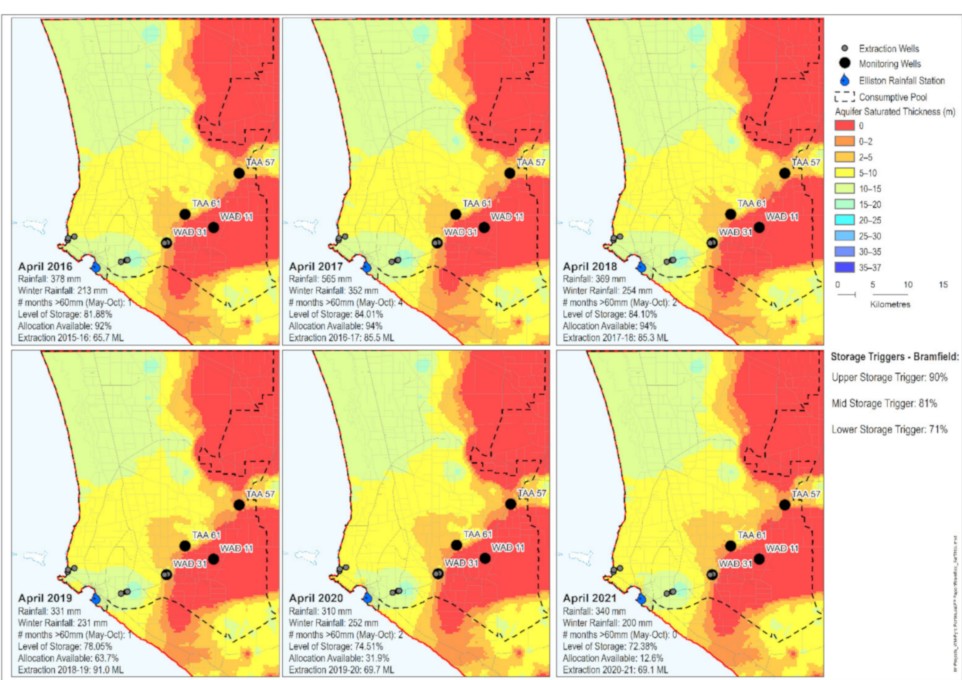

**Figure 8.** Saturated thickness, rainfall statistics, level of storage, percentage of allocation available and extraction for the Bramfield consumptive pool for 2016 to 2021.

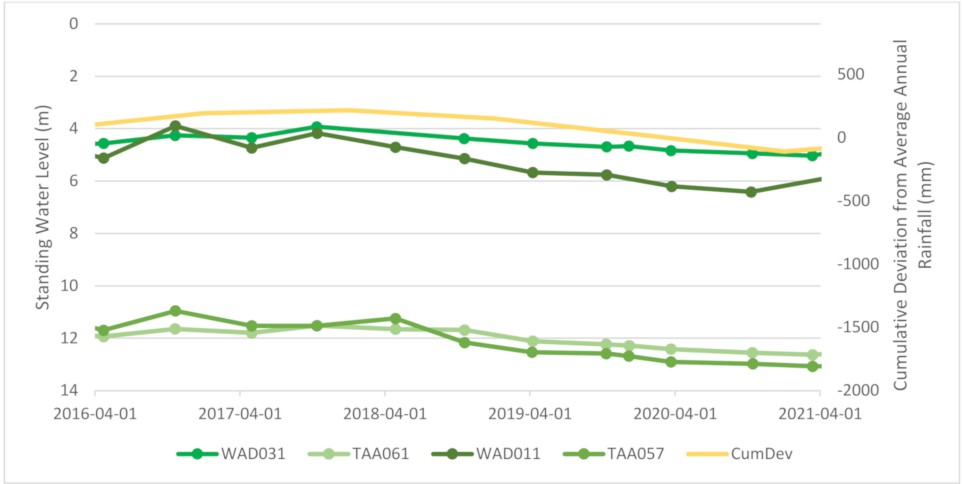

**Figure 9.** Hydrographs for monitoring wells identified in Figure 8 within the Bramfield consumptive pool.

## 5. Discussion

The key role of the water allocation plan, and therefore the outcomes of the approach outlined above, is to ensure that the resource is managed in a sustainable manner which meets the needs of the users, from both a social and economic perspective, alongside protecting the environment which is dependent on the resource.

Monitoring of the key Red Gum communities within the prescribed wells areas at both control sites and sites near extractions indicate that there has been no negative impact of licensed groundwater extraction on the Red Gum groundwater dependent ecosystems between 2016 (commencement of the water allocation plan) until 2020 when reporting occurred [36]. While there are many factors to be considered in the health of groundwater dependent ecosystems, this is a positive indication that despite declines in recharge to

the aquifer occurring over this time, the management response in place is appropriate in supporting the health of the Red Gums.

While this provides some insight into how the approach outlined in this paper with regards to allocating water in the water allocation plan is responding to the declining water availability, a comprehensive review into the effectiveness of the water allocation plan is yet to occur.

Under the legislation, a water allocation plan is required to be comprehensively reviewed at least once in every 10 years to ensure the policies remain fit for purpose, to respond to any new or emerging water management issues, or to update or reflect any new science that may have been undertaken within the region over the life of the water allocation plan.

The aim of the review was to determine whether the principles in the water allocation plan (the policy approach) have been successful in achieving the overarching objectives of the water allocation plan. A successful review is dependent on having collected appropriate monitoring data assessing hydrogeological, ecological and water use parameters throughout the life of the water allocation plan such that when the review takes place it is based on robust information in order to evaluate the success and appropriateness of the water allocation plan.

The comprehensive review for the Southern Basins and Musgrave Prescribed Wells Areas Water Allocation Plan is not due until 2026. As such, a thorough review of the approach to determining allocations presented in this paper has not yet been undertaken. A "light touch" mid-term evaluation of the water allocation plan was recently undertaken, the outcomes of which are mostly focused around providing more frequent and comprehensive reporting on the water resources and the ecosystems dependent on them to the wider community. The mid-term evaluation did not identify any failures in the in the current approach to allocating water and proposes that the approach remains in place until the comprehensive review occurs.

While this adaptive management approach appears to have been successful in restricting water allocations in periods when the resource has inadequate supply, there are some considerations which could be taken into account when the water allocation plan for this region is comprehensively reviewed:

1.  The coastal resources (Coffin Bay and Lincoln South) are buffered by the ocean, and as such the groundwater levels do not decline in response to declining recharge, because in these instances the resource infills with sea water. The freshwater component is possibly diminishing; however, this is not being identified through the use of groundwater level triggers alone. Salinity triggers or freshwater thickness assessments would be beneficial for these particular resources. This limitation was identified in the development of the water allocation plan, but there was insufficient information available at the time to develop suitable salinity triggers.

2.  While some resources were set aside solely for public water supply, other resources which are used for potable supplies in addition to other purposes were not reserved for this purpose. As groundwater levels continue to decline it is likely that water availability for potable demand may not be available from the groundwater resource under the provisions of the water allocation plan. In such instances, consideration could be given to elevating the security of water access entitlements issued for public water supply such that when allocations are reduced at lower levels, these allocations, required to meet critical human water needs, are maintained at a level sufficient to meet the community's minimum requirements, at least for a specified period of time, to ensure alternative water supply schemes such as desalination have sufficient time to be commissioned.

3.  The change in recharge to the aquifer, particularly in the past 5 years while the water allocation plan has been in place, has resulted in groundwater levels declining beyond those observed during the Millennium Drought. While the triggers took into account the presence of groundwater dependent ecosystems, and in many cases-based

the lower storage trigger at the point which aligned to a period during the Millennium Drought at which groundwater levels were their lowest but through which the groundwater dependent ecosystems were maintained, a priority of use scenario was not considered for periods in which the groundwater dependent ecosystems would be at risk despite allocations being ceased. The benefit of reducing allocations in response to a declining groundwater level is that it slows the rate of loss of water supply to the ecosystem, possibly giving them more time to adapt in addition to providing additional time for alternative water sources to come on line. This may enable the groundwater dependent ecosystems to survive through the period of relative shortage to a time in the future when there is less demand on the water resource. However, some consideration should be given to resurveying the key groundwater dependent ecosystems within the prescribed wells areas to have a clear understanding of their reliance on the groundwater resource, if they are no longer reliant on the groundwater resource (and haven't been for some time but are still surviving), then it may be possible to reduce the lower storage trigger if it is no longer required to sustain flow of groundwater to the ecosystem. This may result in social benefits, such as licensees of permanent plantings being able to continue to take a portion of water if that water is no longer required by the groundwater dependent ecosystem.

## 6. Conclusions

Through incorporating these changes to water management arrangements in the legislation into the water allocation plan, the resource is now able to be managed annually to adapt to the resource condition at a particular point in time relative to a baseline. This has only been possible due to amendments to legislation that have allowed a water allocation method to be developed with the flexibility to respond to the observed conditions of the groundwater resources in light of a thorough technical understanding of their structure and dynamics, without impacting on users water rights. The resource managers are also able to monitor the changes over time and determine an appropriate time in which to intervene and propose additional management strategies as the resource becomes stressed (such as an early review of the water allocation plan). This is especially important in regional areas such the Eyre Peninsula, where groundwater level response is highly correlated with climate variability in a drying climate.

**Author Contributions:** Conceptualization, S.S.; methodology, S.S.; validation, G.G.; formal analysis, S.S.; investigation, S.S.; resources, S.S. and G.G.; data curation, S.S.; writing—original draft preparation, S.S. and G.G.; writing—review and editing, S.S. and G.G.; visualization, S.S.; supervision, G.G.; project administration, S.S.; funding acquisition, N/A. All authors have read and agreed to the published version of the manuscript.

**Funding:** Funding for this project was provided by the then Eyre Peninsula Natural Resources Management Board.

**Institutional Review Board Statement:** Not applicable.

**Informed Consent Statement:** Not applicable.

**Data Availability Statement:** Data outputs are not currently available publically but can be requested through the Department for Environment and Water.

**Acknowledgments:** Darren Alcoe, Lyz Risby and Steve Barnett provided assistance with the conceptualisation, data provision and interpretation for this paper.

**Conflicts of Interest:** The authors declare no conflict of interest.

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
