# Peer review of "The Importance of Legislative Reform to Enable Adaptive Management of Water Resources in a Drying Climate"

_water, doi:10.3390/w14091404_

Round 1

Reviewer 1 Report

see attachment

Author Response

Thank you for your feedback, please see the attachment for a collation of all the reviewers comments 

Reviewer 2 Report

The topic is interesting and important.  However, the paper needs much improvements before it can be accepted for publication. 

  1. Literature review part is quite poor.There are quite a bit of similar studies, which needs proper review and analysis and need to provide the uniqueness of the current paper. 
  2. The data presented in Figure 2 and 3 are quite old.Not sure why authors are using 2011 data to address 2022 problems. The same for figure 3, which is 2015 data. 
  3. I strongly suggest the authors to look at the IPCC working group recent report.
  4. It is not clear how the values in Table 1 (trigger level) were derived.
  5. Discussion part is more descriptive.It needs more analytical inputs with reference to previous work. 

Author Response

(The authors gave the same response as above.)

Reviewer 3 Report

This is an interesting paper and certainly appropriate for the special issue. Couple of minor points - page 4 water table is normally two word. Discussion item 3, page 12 (and elsewhere) - one needs to be careful about statements "change in climate in just the past 5 years" has resulted in many low rainfall years. That may very well be true, but 5 years with lower rainfall could be normal climatic variation.

Author Response

(The authors gave the same response as above.)

Reviewer 4 Report

The proposed paper is relevant for the readers and very well written. 

It presents an interesting approach on groundwater management, based on sound studies and valuable experience. 

Author Response

(The authors gave the same response as above.)

Round 2

Reviewer 2 Report

I think the paper is substantially improved, and it can be accepted.